# CASteer: Cross-Attention Steering for Controllable Concept Erasure

**Tatiana Gaintseva[1,2], Andreea-Maria Oncescu[2], Chengcheng Ma[3], Ziquan Liu[1], Martin Benning[4], Gregory Slabaugh[1], Jiankang Deng[2,5], Ismail Elezi[2]**

[1]Queen Mary University of London  [2]Huawei Noah's Ark  [3]CASIA
[4]University College London  [5]Imperial College London

## Abstract

Diffusion models have transformed image generation, yet controlling their outputs to reliably erase undesired concepts remains challenging. Existing approaches usually require task-specific training and struggle to generalize across both concrete (e.g., objects) and abstract (e.g., styles) concepts. We propose CASteer (**C**ross-**A**ttention **Steer**ing), a training-free framework for concept erasure in diffusion models using steering vectors to influence hidden representations dynamically. CASteer precomputes concept-specific steering vectors by averaging neural activations from images generated for each target concept. During inference, it dynamically applies these vectors to suppress undesired concepts only when they appear, ensuring that unrelated regions remain unaffected. This selective activation enables precise, context-aware erasure without degrading overall image quality. This approach achieves effective removal of harmful or unwanted content across a wide range of visual concepts, all without model retraining. CASteer outperforms state-of-the-art concept erasure techniques while preserving unrelated content and minimizing unintended effects. Code is available at https://github.com/Atmyre/CASteer.

## 1 Introduction

Recent advances in diffusion models Ho et al. (2020); Rombach et al. (2022) have revolutionized image Podell et al. (2024) and video generation Girdhar et al. (2024), achieving unprecedented realism. These models operate by gradually adding noise to data during a forward process and then learning to reverse this noise through a series of iterative steps, reconstructing the original data from randomness. By leveraging this denoising process, diffusion models generate high-quality, realistic outputs, making them a powerful tool for creative and generative tasks.

However, the same capabilities that make diffusion models transformative also raise profound ethical and practical concerns. The ability to generate hyper-realistic content amplifies societal vulnerabilities. Risks range from deepfakes and misinformation to subtler effects such as erosion of trust in digital media and targeted manipulation. Addressing these challenges requires not only reactive safeguards (e.g., blocking explicit content) but proactive methods to constrain or remove harmful concepts at the level of the model itself. Current approaches to moderation often treat symptoms rather than causes, limiting their adaptability as risks and applications evolve.

Existing methods for concept erasure in diffusion models remain narrow in scope. LoRA-based fine-tuning Hu et al. (2022) is effective for removing specific objects or styles but struggles with abstract or composite concepts (e.g., nudity, violence, or ideological symbolism), and scales poorly when multiple concepts must be removed, requiring separate adapters or costly retraining. Prompt-based interventions Yoon et al. (2024) offer greater flexibility for abstract harm reduction but lack precision in suppressing concrete attributes, often failing to generalize across concept variations. As a result, existing strategies fall short of delivering reliable, efficient, and broad-spectrum concept erasure.

In this work, we introduce CASteer, a training-free method for controllable concept erasure that leverages the principle of *steering* to influence hidden representations of diffusion models dynamically. Our method builds on recent findings that deep neural networks encode features into approximately

linear subspaces Elhage et al. (2021); Wu et al. (2023). Prior research has shown that intermediate subspaces of diffusion backbones also exhibit this property, with directions that modulate the strength of particular features Kwon et al. (2023); Park et al. (2023); Si et al. (2024); Tumanyan et al. (2023); Li et al. (2024). Yet, these techniques remain limited in scope, often restricted to specific subspaces, requiring training, or offering only coarse control.

Our approach departs from this paradigm. We show that *multiple* subspaces within diffusion models exhibit linear properties that can be harnessed for precise concept erasure. For each concept of interest, we generate $k$ *positive* images (where $k \geq 1$) containing the concept and $k$ *negative* images not containing it, and compute the steering vectors by subtracting the averaged hidden representations of the network across *negative* images from those of *positive* ones. During inference, these precomputed vectors are applied directly to the model activations, allowing us to selectively suppress undesirable concepts without retraining or degrading the overall image quality. Experiments demonstrate that CASteer achieves fine-grained erasure of harmful or unwanted concepts (e.g., nudity, violence), while maintaining robustness across a wide range of diffusion models, including SD 1.4, SDXL Podell et al. (2024), Sana Xie et al. (2025), and their distilled variants (e.g., SDXL-Turbo Sauer et al. (2022), Sana-Sprint Chen et al. (2025)).

In summary, our **contributions** are the following:

- We **propose** a novel training-free framework for controllable concept erasure in diffusion models, leveraging steering vectors to suppress unwanted image features without retraining.
- We **demonstrate** that CASteer effectively handles both concrete (e.g., specific characters) and abstract (e.g., nudity, violence) concepts, and scales to multiple simultaneous erasures.
- We **achieve** state-of-the-art performance in concept erasure across diverse tasks and diffusion backbones, validating the robustness, versatility, and practicality of our approach.

## 2 RELATED WORK

**Data-driven AI Safety.** Ensuring the safety of image and text-to-image generative models hinges on preventing the generation of harmful or unwanted content. Common approaches include curating training data with licensed material Rao (2023); Schuhmann et al. (2022), fine-tuning models to suppress harmful outputs Rombach et al. (2022); Shi et al. (2020), or deploying post-hoc content detectors Bedapudi (2022); Rando et al. (2022). While promising, these strategies face critical limitations: data filtering introduces inherent biases Shi et al. (2020), detectors are computationally efficient but often inaccurate or easily bypassed Gandikota et al. (2023); SmithMano (2022), and model retraining becomes costly when new harmful concepts emerge. Alternative methods leverage text-domain interventions, such as prompt engineering Shi et al. (2020) or negative prompts Miyake et al. (2023); Schramowski et al. (2023). Yet these remain vulnerable to adversarial attacks, lack flexibility and precision as they operate in the discrete space of tokens, and often fail to address the disconnect between text prompts and visual outputs—models can still generate undesired content even when text guidance is "safe". Our approach instead operates in the joint image-text latent space of diffusion models, enabling more robust and granular control over generated content without relying solely on textual constraints.

**Model-driven AI Safety.** Current methods Gandikota et al. (2023); Kumari et al. (2023); Heng & Soh (2023); Zhang et al. (2024a); Huang et al. (2024); Lee et al. (2025) erase unwanted concepts by fine-tuning or otherwise optimising models and adapters to shift probability distributions toward null or surrogate tokens, often combined with regularization or generative replay Shin et al. (2017). Other methods, such as Gandikota et al. (2024); Gong et al. (2024), use direct weight editing to remove unwanted concepts. Although effective, these approaches lack precision, inadvertently altering or removing unrelated concepts. Advanced techniques like SPM Lyu et al. (2024) and MACE Lu et al. (2024) improve specificity through LoRA adapters Hu et al. (2022), transport mechanisms, or prompt-guided projections to preserve model integrity. However, while promising for concrete concepts (e.g., Mickey Mouse), they still struggle with abstract concepts (e.g., nudity) and require parameter updates. Another group of methods focuses on interventions into internal mechanisms of generative models. Methods like Prompt-to-Prompt Hertz et al. (2023) enable fine-grained control over text-specified concepts (e.g., amplifying or replacing elements) through interventions to cross-attention maps, yet fail to fully suppress undesired content, particularly when concepts are implicit or absent from prompts. This task-specific specialization limits their utility for safety-critical

erasure, where complete removal is required. CASteer bridges this gap, enabling precise, universal concept suppression without relying on textual priors or compromising unrelated model capabilities. Another area of research focuses on removing information about undesired concepts from text embeddings that generative models are conditioned on Yoon et al. (2024); Zhang et al. (2024b); Qiu et al. (2024). However, as these methods operate on a discrete space of token embeddings, their trade-off between the effectiveness of erasure and the preservation of other features is limited. Zhang et al. (2024c) proposes using adversarial training for concept unlearning; however, training this method is computationally intensive. In contrast, CASteer eliminates training entirely, enabling direct, non-invasive concept suppression in the model's latent space without collateral damage to unrelated features.

**Utilizing directions in latent spaces.** This area of research focuses on finding interpretable directions in various intermediate spaces of diffusion models Kwon et al. (2023); Park et al. (2023); Si et al. (2024); Tumanyan et al. (2023), which can then be used to control the semantics of generated images. Based on this idea, SDID Li et al. (2024) recently proposed to learn a vector for each given concept, which is then added to the intermediate activation of a bottleneck layer of the diffusion model during inference to provoke the presence of this concept in the generated image. However, this method is highly architecture-specific and fails to deliver precise control over attributes. In our work, we propose a training-free method for constructing interpretable directions in intermediate activation spaces of various diffusion models for more precise control of image generation. SAeUron Cywiński & Deja (2025) utilizes Sparse Autoencoders Olshausen & Field (1997) (SAEs) to find interpretable directions in the activation space of the diffusion model. However, SAEs are unstable, require extensive training, and do not provide initial control over the set of attributes that can be erased. In contrast, CASteer does not require training and provides direct control over the manipulated attributes.

## 3 METHODOLOGY

The main operating principle of CASteer is to modify outputs of certain intermediate layers during inference in order to affect the semantics of generated images, thus preventing the generation of a desired concept. These outputs are modified using specially designed *steering vectors*. In this section, we begin by justifying the choice of the intermediate layers that CASteer modifies (Sec. 3.1), then proceed with the procedure of construction of steering vectors (Sec. 3.2), and after that describe how these steering vectors are used during inference to control the generation process (Sec. 3.3). Finally, we elaborate on practical aspects regarding the calculation and use of the steering vectors (Sec. 3.4).

### 3.1 CHOICE OF LAYERS TO STEER

Most modern diffusion models use U-Net or Diffusion Transformers (DiT) Peebles & Xie (2023) as a backbone. They consist of a set of Transformer blocks, each having three main components: cross-attention (CA) layer, self-attention (SA) layer, and MLP layer, all of which contribute to the residual stream of the model. Among those, CA layers are the only place in the model where information from the text prompt goes into the model, guiding text-to-image generation. For every image patch and prompt embedding, each CA layer generates a vector matching the size of the image patch embedding. After summation, these vectors transmit text-prompt information to corresponding image regions Hertz et al. (2023).

As the semantics of the resulting image is mostly determined by the text prompt, we modify the outputs of the CA layers during inference, which results in effective, yet precise, control over the features of the generated image. Thus, CASteer constructs steering vectors for the outputs of every CA layer in the model. In the appendix, we present experiments on applying CASteer to steer outputs of other layers (SA, MLP, and outputs of intermediate layers inside CA blocks).

### 3.2 CONSTRUCTION OF STEERING VECTORS

We propose to construct steering vectors for each concept we aim to manipulate. These vectors correspond to the cross-attention (CA) outputs we modify. Each steering vector matches the size of the CA outputs and encodes the desired concept's information. For preventing the concept from being present in the generated image, we subtract steering vectors of an unwanted concept from cross-attention outputs during generation.

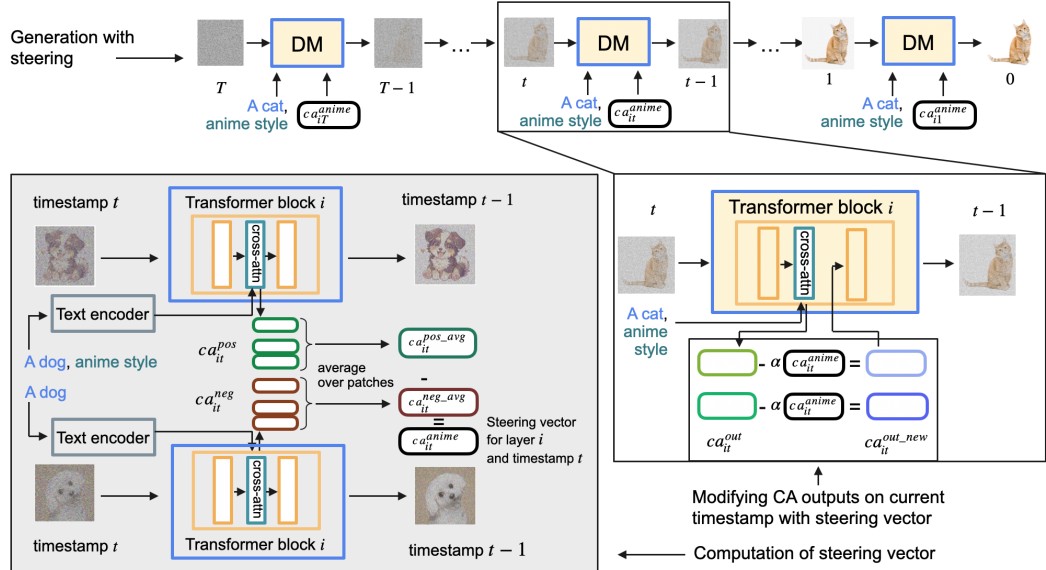

Figure 1: Main pipeline. (Bottom left, gray background) For computing a steering vector, we prompt diffusion model with two prompts that differ in a desired concept, e.g., "anime style" and save CA outputs at each timestamp $t$ and each CA layer $i$. We average these outputs over image patches and get averaged CA outputs $ca_{it}^{pos\_avg}$ and $ca_{it}^{neg\_avg}$ for each $t$ and $i$. We subtract the latter from the former, getting a steering vector for the layer $i$ and timestamp $t$ $ca_{it}^{anime}$. (Right) For deleting concept $X$ from generation, at each denoising step $t$, we subtract steering vector $ca_{it}^{anime}$ multiplied by intensity $\alpha$ from the CA outputs of the layer $i$.

We construct steering vectors as follows. Given a concept $X$ to manipulate, we create paired positive and negative prompts differing only by the inclusion of $X$. For example, if $X$ = "baroque style", example prompts are $p_{pos}$ = "A picture of a man, baroque style" and $p_{neg}$ = "A picture of a man". Assume a DiT backbone has $N$ Transformer blocks, each containing one CA layer, totaling $N$ CA layers. We generate images from both prompts, saving outputs from each of the $N$ cross-attention layers across all $T$ denoising steps. This yields $NT$ cross-attention output pairs $\langle ca_{it}^{pos}, ca_{it}^{neg} \rangle$ for $1 \leqslant i \leqslant N$ and $1 \leqslant t \leqslant T$, where $i$ denotes the layer and $t$ is the denoising step. Each $ca_{it}^{pos}$ and $ca_{it}^{neg}$ has dimensions patch_num$_i$ × emb_size$_i$, corresponding to the number of patches and embedding size at layer $i$. We average $ca_{it}^{pos}$ and $ca_{it}^{neg}$ over image patches to obtain averaged cross-attention outputs:

$$ca_{it}^{pos\_avg} = \frac{\sum_{k=1}^{patch\_num_i} ca_{itk}^{pos}}{patch\_num_i} \; ; \; ca_{it}^{neg\_avg} = \frac{\sum_{k=1}^{patch\_num_i} ca_{itk}^{neg}}{patch\_num_i} \qquad (1)$$

where $ca_{it}^{pos\_avg}$ and $ca_{it}^{neg\_avg}$ are vectors of size emb_size$_i$. Then, for each of these $N$ layers and each of $T$ denoising steps, we construct a corresponding steering vector carrying a notion of $X$ by subtracting its averaged cross-attention output that corresponds to the negative prompt from that corresponding to the positive one as:

$$ca_{it}^{X} = f_{norm}(ca_{it}^{pos\_avg} - ca_{it}^{neg\_avg}). \qquad (2)$$

where $f_{norm}$ is an $L_2$-normalization function: $f_{norm}(v) = \frac{v}{||v||_2^2}$.

### 3.3 USING STEERING VECTORS TO CONTROL GENERATION

Computed steering vectors can be seen as directions in a space of intermediate representations of a model (in the space of CA activations) that represent a notion of $X$. Thus, we should be able to control the expressiveness of certain feature $X$ by steering the model representations along the steering vector representing $X$. That is, we can prevent a concept from appearing on the generated image by subtracting some amount of steering vectors for that concept from corresponding CA outputs of a model during inference:

$$ca_{itk}^{out\_new} = ca_{itk}^{out} - \alpha ca_{itk}^{X}, \qquad (3)$$

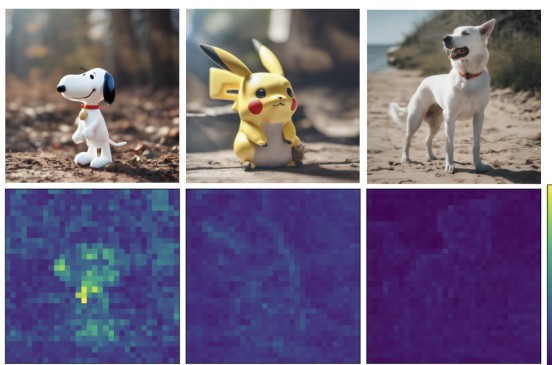

Figure 2: Heatmaps of dot product values between CA outputs of SDXL model (layer 15, denoising step 0) and steering vector for Snoopy concept". Top: generated images, bottom: heatmaps. Images are generated based on the prompt "A bright photo of a X", where X ∈ Snoopy", Pikachu", dog". We see that for the image that contains Snoopy (left), dot product values are high for those image tokens that correspond to the image tokens that actually produce Snoopy. For images that do not contain Snoopy, dot product values are low for all image tokens.

Here $1 \leqslant k \leqslant$ patch_num$_i$, and $\alpha$ is a hyperparameter that controls the strength of concept suppression. Larger values of $\alpha$ lead to higher suppression of the concept $X$ in the generated image. Below we propose a way to adjust $\alpha$ dynamically based on activations of diffusion model during generation, achieving effective and precise erasure of unwanted concepts in the resulting image.

**Choice of alpha.** Most often when we aim to suppress the concept of $X$, our goal is to completely prevent it from appearing on any generated image given any input prompt. This is the case of such tasks as nudity/violence removal or privacy, when we do not want the model to ever generate somebody's face or artwork. However, there might be different magnitudes for concept $X$ in the original text prompt (e.g., prompts "an angry man" or "a furious man" express different levels of anger). A concept $X$ can have different magnitudes of expression in different patches of the image being generated. Consequently, if we use Eq. 3 for suppression, different values of $\alpha$ are needed to completely suppress $X$ for different prompts and individual image patches while not affecting other features in the image.

We propose to estimate $\alpha$ for concept deletion by using the dot product between $ca_{it}^X$ and corresponding CA output $ca_{itk}^{out}$ ($\langle ca_{it}^X, ca_{itk}^{out} \rangle$) as an assessment of amount of $X$ that is present in the image part corresponding to $k^{th}$ patch of $ca_{it}^{out}$ (see Fig. 2). As $ca_{it}^X$ is normalized, the value of this dot product is the length of the projection of the CA output $ca_{itk}^{out}$ onto the steering vector $ca_{it}^X$. As $ca_{it}^X$ can be seen as a direction in a linear subspace corresponding to the concept $X$, the length of the projection can be seen as the amount of $X$ that is present in $ca_{itk}^{out}$. That said, for removing information about $X$ from $ca_{itk}^{out}$, we propose to subtract the amount of $ca_{it}^X$ proportionate to the dot product between $ca_{it}^X$ and $ca_{itk}^{out}$ from $ca_{itk}^{out\_new}$, i.e., define $\alpha = \beta(ca_{it}^X, ca_{itk}^{out})$. Consequently, Eq. 3 becomes the following:

$$ca_{itk}^{out\_new} = ca_{itk}^{out} - \beta \langle ca_{it}^X, ca_{itk}^{out} \rangle ca_{it}^X. \tag{4}$$

Here $1 \leqslant k \leqslant$ patch_num$_i$, and $\beta$ is a hyperparameter that controls the strength of the suppression.

Note that Eq. 4 can be reformulated in a matrix form as a projection operator onto the subspace orthogonal to steering vector $s = ca_{it}^X$:

$$s^{new} = f_{\text{delete}}(c, s) = (I - \beta s s^T) c \tag{5}$$

Here $s^{new} = ca_{itk}^{out\_new}$, $c = ca_{itk}^{out}$, $s = ca_{it}^X$, $I$ is an identity matrix.

**Intermediate clipping**. We now introduce a mechanism of clipping the value of $\alpha$ to get better control over concept suppression. Note that using Eq. 4 we only want to influence those CA outputs $ca_{itk}^{out}$ which have a positive amount of unwanted concept $X$. As dot product $\langle ca_{it}^X, ca_{itk}^{out} \rangle$ measures the amount of $X$ present in CA output $ca_{itk}^{out}$, we only want to steer those CA outputs $ca_{itk}^{out}$, which have a positive dot product with $ca_{it}^X$. So the equation becomes the following:

$$\alpha = \max(\beta \langle ca_{it}^X, ca_{itk}^{out} \rangle, 0)$$
$$ca_{itk}^{out\_new} = ca_{itk}^{out} - \alpha ca_{it}^X. \tag{6}$$

Note that if intermediate clipping is used, Eq. 6 can no longer be formulated in a matrix form. In the experiments section, we present results of applying CASteer for concept erasure both with and without intermediate clipping (i.e. using Eq. 4 and Eq. 6).

### 3.4 PRACTICAL CONSIDERATIONS

**Multiple Prompts for Steering Vector.** We described in the previous section how to construct and use steering vectors to alter one concept, based on one pair of prompts, e.g., "a picture of a man" and "a picture of a man, baroque style". As mentioned, a steering vector can be seen as the direction in the space of intermediate representations of a model that points from an area of embeddings not containing a concept $X$, to an area that contains it. In order for this direction to be more precise, we propose to construct steering vectors based on multiple pairs of prompts instead of one. More precisely, we obtain $P \geqslant 1$ pairs of $ca_{itp}^{pos\_avg}$ and $ca_{itp}^{neg\_avg}$, $1 \leqslant p \leqslant P$, then average them over P:

$$ca_{it}^{pos\_avg} = \frac{\sum_{p=1}^{P} ca_{itp}^{pos\_avg}}{P}, ca_{it}^{neg\_avg} = \frac{\sum_{p=1}^{P} ca_{itp}^{neg\_avg}}{P} \tag{7}$$

and obtain steering vectors as $ca_{it}^{X} = ca_{it}^{pos\_avg} - ca_{it}^{neg\_avg}$.

**Steering multiple concepts.** It is easy to erase multiple concepts during a generation by either mutually orthogonalizing a set of steering vectors corresponding to these concepts and applying them to the cross-attention output successively, or constructing single steering vector corresponding to multiple concepts by simply averaging individual steering vectors. In the experimental section, we show results on applying a steering vector constructed for multiple concepts using averaging approach to prevent generation of inappropriate concepts. Additionally, in the appendix (Sec. M) we show results on erasing multiple concepts by orthogonalizing their corresponding steering vectors.

**Efficiency: Transferring vectors from distilled models.** Adversarial Diffusion Distillation (ADD) Sauer et al. (2022) is a fine-tuning approach that allows sampling large-scale foundational image diffusion models in $1$ to $4$ steps, while producing high-quality images, with many methods such as SDXL and Sana having distilled versions (SDXL-Turbo and Sana-Sprint). We observe that steering vectors obtained from the distilled models can successfully be used for steering generations of its corresponding non-distilled variants. More formally, having a pair of prompts, we obtain $ca_i^{pos\_avg}$ and $ca_i^{neg\_avg}$ from the distilled model using 1 denoising step. Note that there is no second index $t$ as we use only one denoising iteration, i.e. $T = 1$. We then construct steering vectors for the concept $X$ as $ca_i^{X} = ca_i^{pos\_avg} - ca_i^{neg\_avg}$ and then use it to steer non-Turbo variant of the model by using $ca_i^{X}$ for each denoising step $1 \leqslant j \leqslant T$.

**Injecting CASteer into model weights.** Note that when steering more advanced models (SDXL or Sana), we use steering vectors from Turbo/Sprint model versions, where we have only one steering vector per model CA layer. Also note that the last layer of CA block in SDXL/SANA is Linear layer with no bias and no activation function, i.e., essentially is a matrix multiplication: $h_{out} = W_{proj\_out} h_{in}$. Here $W_{proj\_out}$ is a weight matrix of the last $proj\_out$ layer of CA block of SDXL/SANA, $h_{in}$ and $h_{out}$ are input and output to that layer, $h_{out}$ being the final output of CA layer. In this case, by combining last layer of CA block with CASteer formulation in a matrix form (Eq. 5), we can incorporate CASteer directly into weights of the model, by multiplying weight matrix of the last layer of CA block with $I - ss^T$ matrix from Eq. 5:

$$h_{out} = (I - ss^T) W_{proj\_out} h_{in} = W_{proj\_out}^s h_{in} \tag{8}$$

$W_{proj\_out}^s$ is a matrix of the same size as $W_{proj\_out}$. This results in having zero inference overhead compares to original SDXL/SANA model similar to LoRA-like tuning approaches.

## 4 EXPERIMENTS

We evaluate the performance of our method on the task of erasing different concepts. We show that our method succeeds in suppressing both abstract (e.g., "nudity", "violence") and concrete concepts (e.g., "Snoopy"). Moreover, we demonstrate the advantages of our method in removing implicitly defined concrete concepts (e.g., if a concept is "Mickey", prompting "a mouse from a Disneyland" should not result in a generation of Mickey).

**Implementation details.** For a fair comparison, we report our main quantitative results using StableDiffusion-v1.4 (SD-v1.4) Rombach et al. (2022). SD-1.4 model does not have a Turbo version, so for these experiments we use per-step steering vectors computed from the original SD-1.4. We apply steering to all of the CA layers in the model. We set $\beta = 2$ for the concept erasure in all

Table 1: **Quantitative results on nudity removal based on I2P Schramowski et al. (2023) dataset.** Detection of nude body parts is done by Nudenet at a threshold of 0.6. F: Female, M: Male. The best results are highlighted in bold, second-best are underlined.

| Method | Nudity Detection | | | | | | | | |
|---|---|---|---|---|---|---|---|---|---|
| | Breast(F) | Genitalia(F) | Breast(M) | Genitalia(M) | Buttocks | Feet | Belly | Armpits | Total↓ |
| SD v1.4 | 183 | 21 | 46 | 10 | 44 | 42 | 171 | 129 | 646 |
| DoCo Wu et al. (2025) | 162 | 29 | 48 | 63 | 64 | 122 | 168 | 250 | 906 |
| Ablating (CA) Kumari et al. (2023) | 298 | 22 | 67 | 7 | 45 | 66 | 180 | 153 | 838 |
| FMN Zhang et al. (2024a) | 155 | 17 | 19 | 2 | 12 | 59 | 117 | 43 | 424 |
| ESD-x Gandikota et al. (2023) | 101 | 6 | 16 | 10 | 12 | 37 | 77 | 53 | 312 |
| SLD-Med Schramowski et al. (2023) | 39 | 1 | 26 | 3 | 3 | 21 | 72 | 47 | 212 |
| UCE Gandikota et al. (2024) | 35 | 5 | 11 | 4 | 7 | 29 | 62 | 29 | 182 |
| SA Heng & Soh (2023) | 39 | 9 | 4 | **0** | 15 | 32 | 49 | 15 | 163 |
| ESD-u Gandikota et al. (2023) | 14 | 1 | 8 | 5 | 5 | 24 | 31 | 33 | 121 |
| Receler Huang et al. (2024) | 13 | 1 | 12 | 9 | 5 | 10 | 26 | 39 | 115 |
| MACE Lu et al. (2024) | 16 | **0** | 9 | 7 | 2 | 39 | 19 | 17 | 109 |
| RECE Gong et al. (2024) | 8 | **0** | 6 | 4 | **0** | 8 | 23 | 17 | 66 |
| CPE (one word) Lee et al. (2025) | 11 | 2 | 3 | 2 | 5 | 15 | 13 | 15 | 66 |
| CPE (four word) Lee et al. (2025) | 6 | 1 | 3 | 2 | 2 | 8 | 8 | 10 | 40 |
| AdvUnlearn Zhang et al. (2024c) | **1** | 1 | **0** | **0** | **0** | 13 | **0** | 8 | 23 |
| SAeUron Cywiński & Deja (2025) | 4 | **0** | **0** | 1 | 3 | 2 | 1 | 7 | 18 |
| Ours (w/o clip) | 5 | **0** | **0** | 1 | 3 | 2 | **0** | 1 | 12 |
| Ours (clip) | 4 | **0** | **0** | 1 | 2 | **0** | **0** | **0** | **7** |

Table 2: **Quantitative results on inappropriate content removal based on I2P Schramowski et al. (2023) dataset. Detection of inappropriate content is done by Q16 Schramowski et al. (2022) classifier.** The best results are highlighted in bold, second-best are underlined.

| Class name | Inappropriate proportion (%) (↓) | | | | | | | | | |
|---|---|---|---|---|---|---|---|---|---|---|
| | SD | FMN | Ablating | ESD-x | SLD | ESD-u | UCE | Receler | Ours (w/o clip) | Ours (clip) |
| Hate | 44.2 | 37.7 | 40.8 | 34.1 | 22.5 | 26.8 | 36.4 | **28.6** | 35.5 | 29.00 |
| Harassment | 37.5 | 25.0 | 32.9 | 30.2 | 22.1 | 24.0 | 29.5 | **21.7** | 29.85 | 25.61 |
| Violence | 46.3 | 47.8 | 43.3 | 40.5 | 31.8 | 35.1 | 34.1 | **27.1** | 32.54 | 27.78 |
| Self-harm | 47.9 | 46.8 | 47.4 | 36.8 | 30.0 | 33.7 | 30.8 | 24.8 | **26.10** | 26.22 |
| Sexual | 60.2 | 59.1 | 60.3 | 40.2 | 52.4 | 35.0 | 25.5 | 29.4 | 22.99 | **20.73** |
| Shocking | 59.5 | 58.1 | 57.8 | 45.2 | 40.5 | 40.1 | 41.1 | 34.8 | 38.43 | **34.00** |
| Illegal activity | 40.0 | 37.0 | 37.9 | 28.9 | 22.1 | 26.7 | 29.0 | 21.3 | 21.46 | **17.61** |
| Overall | 48.9 | 47.8 | 45.9 | 36.6 | 33.7 | 32.8 | 31.3 | 27.0 | 28.94 | **25.58** |

experiments. This choice is motivated by the fact that with $\beta = 2$, the Eq. 5 becomes a Householder operator (reflection) of the CA activation vector $c$ across the hyperplane orthogonal to the steering vector $s$. This operation preserves $L_2$-norm of the vector $c$, thus keeping relative and absolute values of all the information present in $c$ that is orthogonal to $s$ intact after transformation. In appendix (see Sec. H.3) we ablate the choice of $\beta$. We show $\beta < 2$ leads to lower level of concept suppression, while still producing high quality images, enabling control on the level of concept erasure. We also show that for SDXL and SANA models, values of $\beta > 2$ lead to stronger concept erasure, while also leaving general quality of images high. We use 50 prompt pairs for generating steering vectors for concrete concepts (e.g., "Snoopy"), and 196 prompt pairs for generating steering vectors for abstract concepts (e.g., "nudity"). Information about prompts used for generation of the steering vectors is in the appendix.

We also show effectiveness of CASteer on bigger models, such as SDXL and SANA, and present results on these models in the appendix. For SDXL Podell et al. (2024) and SANA, we use steering vectors obtained from SDXL-Turbo and SANA-Sprint models, respectively.

## 4.1 RESULTS

**Abstract concept erasure.** In this section, we present results on inappropriate content erasure based on I2P dataset Schramowski et al. (2023). I2P is a dataset of 4,703 curated prompts designed to test generative models, where most prompts lead to images containing inappropriate content. Following prior work, we test CASteer on two I2P-based tasks: 1) removing nudity, 2) removing all inappropriate content at once. For nudity removal, we utilize CASteer with steering vectors generated for the concept of "nudity". For inappropriate content removal, we use CASteer with steering vectors obtained as average of steering vectors generated for each type of inappropriate content, i.e., hate, harassment, violence, self-harm, shocking, sexual, and illegal content.

We compare our method with the following state-of-the-art approaches: Ablating (CA) Kumari et al. (2023), FMN Zhang et al. (2024a), SAeUron Cywiński & Deja (2025) SLD Schramowski et al. (2023), ESD Gandikota et al. (2023), UCE Gandikota et al. (2024), MACE Lu et al. (2024), SA Heng & Soh (2023), Receler Huang et al. (2024), RECE Gong et al. (2024), CPE Lee et al. (2025), AdvUn Zhang et al. (2024c) and . Following prior art, we utilize the NudeNet [1] to detect nude body

---
[1] https://github.com/notAI-tech/NudeNet

parts on generated images for nudity erasure task, and the NudeNet with the Q16 detector to detect inappropriate content.

We present the results for CASteer versions with and without intermediate clipping applied in Tab. 1 and Tab. 2. We show that both versions of CASteer outperform all prior models on nudity erasure, with CASteer version with clipping having more than 2 times fewer images with detected nudity than the second-best result. On the inappropriate content removal, CASteer version with clipping also achieves state-of-the-art result, surpassing second-best model Receler by 1.42% overall.

To assess general generation quality of CASteer, we follow prior work and run CASteer with "nudity" steering vectors on prompts from COCO-30k Lin et al. (2014). We report FID Heusel et al. (2017) for general visual quality and CLIP score Hessel et al. (2021) for image-prompt alignment. Results are presented in Tab. 3. Both versions of CASteer have better FID than prior art.

Thus, CASteer clearly is capable of deleting unwanted information while maintaining general high quality. Note that these datasets feature adversarial prompts, i.e., the "nudity" concept is encoded in the prompts implicitly.

Table 3: **Evaluation of nudity-erased models.** Robustness is measured with nudity prompts from the I2P dataset, while locality is assessed using COCO-30K prompts.

| Method | Locality | |
| --- | --- | --- |
| | CLIP-30K($\uparrow$) | FID-30K($\downarrow$) |
| SD v1.4 | 31.34 | 14.04 |
| FMN | 30.39 | 13.52 |
| CA | **31.37** | 16.25 |
| AdvUn | 28.14 | 17.18 |
| Receler | 30.49 | 15.32 |
| MACE | 29.41 | 13.42 |
| CPE | **31.19** | 13.89 |
| UCE | 30.85 | 14.07 |
| SLD-M | 30.90 | 16.34 |
| ESD-x | 30.69 | 14.41 |
| ESD-u | 30.21 | 15.10 |
| SAeUron | 30.89 | 14.37 |
| *Ours (w/o clip)* | 30.69 | 13.28 |
| *Ours (clip)* | 31.09 | **13.02** |

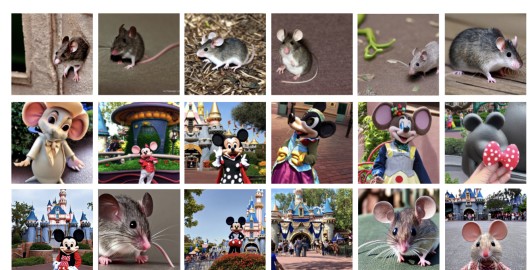

Figure 3: SPM and DoCo failure in removing implicitly defined concepts (SD-1.4). Top: CASteer, Middle: SPM, Bottom: DoCo. Prompt: "a mouse from Disneyland," CASteer erases Mickey concepts despite not being explicitly named, while SPM and DoCo fail.

**Concrete concepts erasure.** To assess ability of CASteer to remove concrete concepts, we follow the experimental setup of SPM Lyu et al. (2024). In this setting, the concept to be erased is *Snoopy*, and images of five additional concepts (*Mickey*, *Spongebob*, *Pikachu*, *dog* and *legislator*) are generated to test the capability of the method to preserve content not related to the concept being removed. The first four of these are specifically chosen to be semantically close to the concept being removed to show the model's ability to perform precise erasure. Following SPM Lyu et al. (2024) and DoCo Wu et al. (2025), we augment each concept using 80 CLIP Radford et al. (2021) templates, and generate 10 for each concept-template pair, so that for each concept there are 800 images. We evaluate the results using two metrics. First, we utilize CLIP Score (CS) Hessel et al. (2021) to confirm the level of the existence of the concept within the generated content. Second, we calculate FID Heusel et al. (2017) scores between the set of original generations of SD-1.4 model and a set of generations of the steered model. We use it to assess how much images of additional (non-Snoopy) concepts generated by the steered model differ from those of generated by the original model. A higher FID value demonstrates more severe generation alteration. We present the results in Tab. 17. In Fig. 4, we also show two types of plots. Fig. 4a pictures normalized clip score of source concept, i.e. "Snoopy" (the lower the better) versus mean normalized clip scores of other concepts (the higher the better). Normalization is done to ensure equal importance of all the concepts in the mean, and done by dividing clip score of images produced by erasing method by clip score of images produced by vanilla SD-1.4. Methods on the left of the plot erase Snoopy well, and methods on top of the plot preserve other concepts well. Fig. 4b pictures normalized clip score of source concept versus mean FID scores of other concepts (the lower the better). Methods on the left of the plot erase Snoopy well, and methods on top of the plot tend not to affect images of other concepts much.

Results show that CASteer maintains good balance between erasing unwanted concept, while preserving other concepts intact. ESD Gandikota et al. (2023) and Receler erase Snoopy well, but also highly affect other concepts, especially related ones such as *Mickey* or *Spongebob*. Note that their CS

Table 4: Comparison of Artist Concept Removal tasks: Famous (left) and Modern artists (right).

| Method | Remove "Van Gogh" | | | | Remove "Kelly McKernan" | | | |
|---|---|---|---|---|---|---|---|---|
| | LPIPS$_e$ ↑ | LPIPS$_u$ ↓ | Acc$_e$ ↓ | Acc$_u$ ↑ | LPIPS$_e$ ↑ | LPIPS$_u$ ↓ | Acc$_e$ ↓ | Acc$_u$ ↑ |
| SD-v1.4 | - | - | 0.95 | 0.95 | - | - | 0.80 | 0.83 |
| CA (Kumari et al., 2023) | 0.30 | 0.13 | 0.65 | 0.90 | 0.22 | 0.17 | 0.50 | 0.76 |
| RECE (Gong et al., 2024) | 0.31 | 0.08 | 0.80 | 0.93 | 0.29 | 0.04 | 0.55 | 0.76 |
| UCE (Gandikota et al., 2024) | 0.25 | **0.05** | 0.95 | **0.98** | 0.25 | **0.03** | 0.80 | **0.81** |
| SLD-Medium (Schramowski et al., 2023) | 0.21 | 0.10 | 0.95 | 0.91 | 0.22 | 0.18 | 0.50 | 0.79 |
| SAFREE (Yoon et al., 2024) | 0.42 | 0.31 | 0.35 | 0.85 | 0.40 | 0.39 | 0.40 | 0.78 |
| Ours (w/o clip) | **0.46** | 0.31 | 0.35 | 0.88 | **0.54** | 0.27 | **0.05** | 0.80 |
| Ours (clip) | 0.44 | **0.30** | **0.25** | 0.86 | **0.54** | 0.28 | **0.05** | **0.81** |

of unrelated concepts (e.g. "Mickey" or "Spongebob") are significantly lower than that of original SD-1.4, indicating that these concepts are also being affected when erasure of "Snoopy" is done. High FID of these methods on these concepts supports this observation. SAFREE shows a reduced level of *Snoopy* erasure compared to that of CASteer, and has lower CS and higher FID on all the other concepts. SPM keeps unrelated concepts almost intact (High CS and low FID), but has a much lower intensity of *Snoopy* erasure. Moreover, SPM fails to erase implicitly defined concepts (see Fig. 3 and Sec. D in the appendix). We provide qualitative results on comparisons in Fig. 5 and in the appendix.

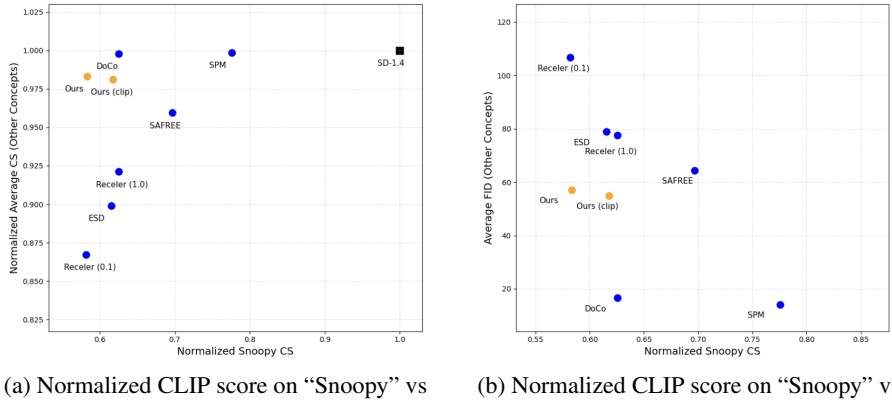

(a) Normalized CLIP score on "Snoopy" vs normalized CLIP scores of other concepts

(b) Normalized CLIP score on "Snoopy" vs FID scores of other concepts

Figure 4: Comparison of various methods on concrete concept erasure (removing "Snoopy")

**Style erasure.** Following SAFREE Yoon et al. (2024) and ESD Gandikota et al. (2023), we also evaluate our method on two **artist-style removal tasks**. One task focuses on the styles of five famous artists (Van Gogh, Picasso, Rembrandt, Warhol, Caravaggio) and the other uses five modern artists (McKernan, Kinkade, Edlin, Eng, Ajin: Demi-Human), with the task being removing the style of Van Gogh and McKernan. Following SAFREE Yoon et al. (2024), we use LPIPS Zhang et al. (2018) and prompt GPT-4o to identify an artist on generated images as evaluation metrics.

We follow SAFREE Yoon et al. (2024) evaluation procedure, please refer to it or our appendix (Sec. E) for more details on the procedure and metrics. From Tab. 4, we see that CASteer achieves the best results in style removal (see columns LPIPS$_e$ and Acc$_e$), while preserving other styles well (see columns LPIPS$_u$ and Acc$_u$). Among all the approaches, CASteer achieves great balance between target style removal and preservation of other styles.

**CASteer is capable of erasing implicitly defined concepts.** We check what happens if we define the prompts implicitly, e.g., "A mouse from Disneyland". We run CASteer, SPM and DoCo trained on the *Mickey* concept on these prompts and show the results in Fig. 3. We clearly see that SPM and DoCo fail to erase the concepts when they are not explicitly defined. In contrast, our method does a much better job of erasing the concepts, despite being implicitly defined. This is also supported by the results on nudity erasure in Tab. 1, as considered datasets contain specially selected adversarial nudity prompts. We provide additional results on implicitly defined prompts in the appendix.

**Overall** experimental results show that CASteer performs precise erasure of both concrete and abstract concepts and concepts defined implicitly while leaving other concepts intact and not affecting the overall quality of generated images. More qualitative results showing the performance of CASteer on prompts related and not related to the target concept can be found in the appendix.

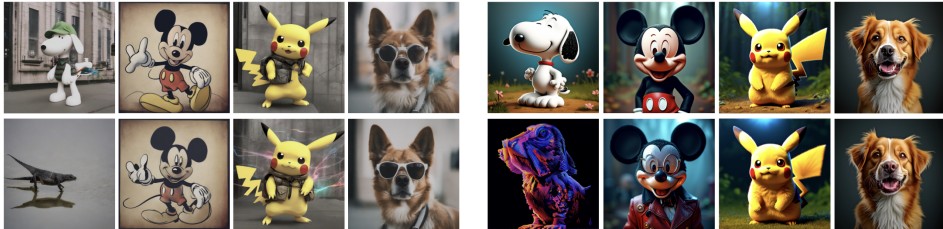

Figure 5: Qualitative results on SDXL (left) and SANA (right) on removing "Snoopy". Top: original model generations, bottom: generations of model steered to remove "Snoopy"

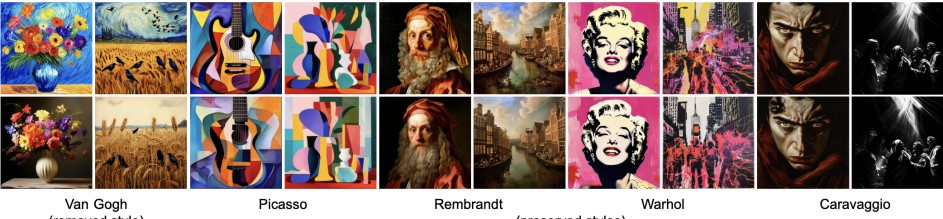

Van Gogh (removed style)     Picasso     Rembrandt    (preserved styles)    Warhol     Caravaggio

Figure 6: Qualitative results on SANA (right) on removing style of "Van Gogh". Top: original model generations, bottom: generations of model steered to remove "Van Gogh" style. Left 2 images correspond to prompts with "Van Gogh style", images on the right correspond to prompts mentioning other artists ( "Picasso", "Rembrandt", "Warhol", "Caravaggio")

## 4.2 ABLATION STUDY

**Steering other layers.** As mentioned in Sec. 3.1, we ablate to determine for which type of layer in the DiT backbone steering is most effective. We show in the appendix that steering the CA outputs is the most effective. We also ablate steering only a fraction of CA layers in Sec. H.5.

**Number of prompt pairs to construct steering vectors.** In the appendix, we provide an ablation on the number of prompt pairs needed to produce high-quality outputs after steering. We find that as little as 50 prompts is enough for the steering vectors to capture the desired concept well.

**Interpretation of steering vectors.** Here, we propose a way to interpret the meaning of steering vectors generated by CASteer. Suppose we have steering vectors generated for a concept $X$ $\{ca_{it}^X\}, 1 \leqslant i \leqslant l, 1 \leqslant t \leqslant T$, where $l$ is the number of model layers and $T$ is the number of denoising steps performed for generating steering vectors. To interpret these vectors, we prompt the diffusion model with a placeholder prompt "X" and at each denoising step, we substitute outputs of the model's CA layers with corresponding steering vectors. This conditions the diffusion model only on the information from the steering vectors, completely suppressing other information from the text prompt. Results are presented in Fig. 41 and in the appendix.

**UMap.** We generate steering vectors for all vocabulary tokens of SDXL text encoders and apply UMap McInnes & Healy (2018) on these steering vectors. We present the results in the appendix, showing that structure emerges in the space of these steering vectors, similar to that of Word2Vec Mikolov et al. (2013), supporting that steering vectors carry the meaning of the desired concept.

**Modern models.** We show qualitative results in SANA and SDXL in Fig. 6. We provide more qualitative and quantitative results in SDXL (Sec. F) and SANA (Sec. G).

**User studies.** In Sec. D, we give several user studies, showing that in most cases, the users prefer our results compared to SPM and Receler.

## 5 CONCLUSION

We presented CASteer, a novel training-free method for controllable concept erasure in diffusion models. CASteer works by using steering vectors in the cross-attention layers of diffusion models. We show that CASteer is general and versatile to work with different versions of diffusion, including distilled models. CASteer reaches state-of-the-art results in concept erasure on different evaluation benchmarks while producing visually pleasing images.

## 6 ACKNOWLEDGMENTS

This work was supported by the Engineering and Physical Sciences Research Council [grant number EP/Y009800/1], through funding from Responsible Ai UK (KP0016).

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
