# OpenReview forum: "CASteer: Cross-Attention Steering for Controllable Concept Erasure"
_ICLR.cc/2026/Conference — ICLR 2026 Poster_

### Official Review · Reviewer_dKH9 · 2025-10-27

**Soundness:** 2
**Presentation:** 2
**Contribution:** 2
**Rating:** 4
**Confidence:** 5

**Summary:**

The paper proposes CASteer, an activation steering method for concept erasure in diffusion models. It extracts steering vectors from cross-attention layers in the diffusion model for the concept to be erased and applies them during inference for erasure. Experiments are conducted for erasure of abstract concepts (nudity, violence) and objects (snoopy, mickey) on I2P benchmark and COCO dataset.

**Strengths:**

The proposed activation steering method for concept unlearning is training-free.

Adapting the method to novel concepts is straightforward and does not require retraining.

Experiments on I2P benchmark for SDv1.4 model shows the effectiveness of the proposed method over a comprehensive set of prior methods.

**Weaknesses:**

The proposed method is not new as it applies the idea of activation steering which is well-established in LLMs to concept unlearning in diffusion models. So, the novelty is limited.

There is no theoretical justification on why operating in the latent space of diffusion models is better. It does not fully justify why unlearning in the cross-attention activation space (latent) is fundamentally better than guidance or prompt-based methods.

Is it sensitive to the noise seed of the diffusion model? Are the steering vectors extracted for random noise seeds?

How does the method work if there is a mismatch between the number of steps used in steering vector extraction and during inference?

Missing ablation study on steering with only one vector per CA layer in SDv1.4.

Concept erasure based on guidance or steering is susceptible to adversarial attacks via concept addition or subtraction. Is this robust to adversarial attacks based on concept arithmetic [1] ?

Results are only reported for few concepts (nudity, violence, snoopy, mickey). Missing erasure results on art styles (e.g., van gogh) or more abstract concepts (e.g., summer, mosaic style etc).

Comparisons are only reported for SDv1.4 model. Missing comparisons to prior methods on more recent models such as SDXL, SANA or SD3.5

Paper can be better organized. Important ablations from supplementary need to be included in the main paper or the appendix instead of supplementary paper.

[1] Petsiuk et. al. Concept Arithmetics for Circumventing Concept Inhibition in Diffusion Models, ECCV 2024

Minor:
Reference to tables in main paper (e.g., table 15) is missing.

**Questions:**

See weaknesses above.

How are the multiple prompts generated?

Do the steering vectors generated for SDv1.4 work for SD v1.5 across different checkpoints of the same model?

---

> ### Author Response · Authors · 2025-11-21
> **Rebuttal to Reviewer dKH9**
>
> We thank the reviewer for the detailed review, and for calling our method **straightforward** while praising its **effectiveness** over a **comprehensive** set of prior methods. Below, we reply to the questions the reviewer raised.
>
> **1. Limited novelty**.
>
> While there is prior work that uses activation steering in LLMs, we are unaware of any prior work that uses non-trainable steering vectors directly to the cross-attention (CA) outputs of diffusion models for concept erasure. Existing approaches that exploit directional structure in diffusion activations (see “Utilizing directions in latent spaces” in Related Work) do not target CA activations. Moreover, many of these methods are architecture-dependent (e.g., requiring access to U-Net bottleneck layers) and training-intensive, limiting applicability to modern non-U-Net backbones such as SANA. Our experiments show that CASteer consistently outperforms these baselines (**Tables 1, 2, 4; Fig. 4**).
>
> We note that the operating principles and backbone structures of diffusion models highly differ from that of LLM, so the transfer of application of steering vectors from LLMs to diffusion models is not straightforward. In our work, we apply steering to the Cross-Attention layers of the model, which is absent in LLMs.
>
> **2. No theoretical justification on why operating in the latent space of diffusion models is better.**
>
> We thank the reviewer for the insightful question. While diffusion models are too complex for a full formal proof comparing all intervention types, we show that under local linearity assumptions and bounded perturbations, steering cross-attention (CA) outputs is provably at least as expressive as steering prompt embeddings.
>
> We added **Theorem L.1 (see Appendix Sec. L)** which states that CA-space interventions can generate perturbations that no shift in text-embedding space can reproduce. Further, the new **Corollary L.1.1** shows that, in the linearized regime, an ideal CA-steering method matches or outperforms any embedding-based method on any concept-erasure objective.
>
> Intuitively, CA outputs lie closer to the model’s residual stream and carry layer-, head-, and spatially localized information about how prompt tokens affect specific regions of the image. Steering here therefore offers more precise and spatially targeted control than global edits to text embeddings.
>
> Empirically, CASteer also outperforms embedding-based approaches, including recent methods such as SAFREE. **We refer to the reviewer to check the newly added Section L (lines 2754-2846) in the Appendix.**
>
> **3. Is it sensitive to the noise seed of the diffusion model? Are the steering vectors extracted for random noise seeds?**
>
> In our main experiments, steering vectors were computed using a fixed seed (seed = 0), while CASteer itself was always applied using varying generation seeds, e.g., the nudity, harmful-content, and style-erasure datasets all provide diverse noise seeds for image generation.
>
> We also assess sensitivity to both prompt choice and random seeds when computing steering vectors in Sec. H.1 (appendix). Using the Snoopy-erasure task with β=2, we recompute steering vectors with different numbers of prompt pairs and different random seeds sampled from the prompt set in Sec. C.1. Each steering vector is evaluated using the Snoopy-erasure protocol (Sec. 4.1), reporting CLIP scores for “Snoopy” and FID for non-target concepts.
>
> **As shown in Fig. 29**, CASteer is robust to the seed used during steering-vector construction, and performance stabilizes for ≥50 prompt pairs.
>
> **4. Missing ablation study on steering with only one vector per CA layer in SDv1.4.**
>
> We thank the reviewer for pointing this out. We note that in SDXL and SANA, we used steering vectors with only one vector per CA layer obtained from distilled versions of these models (SDXL-Turbo and SANA-Sprint, respectively). However, SD-1.4 does not have a corresponding distilled version, thus we used steering vectors from all denoising steps in our main experiments on SD-1.4. However, we added a section in the appendix (Sec. O), showing that for SD-1.4 it is also possible to use one steering vector per CA layer. In this experiment, we use steering vectors for SD-1.4, obtained from the first denoising step. We provide results on all our main experiments (erasing nudity, harmful content, concrete concepts and styles), and demonstrate high performance of CASteer in this setting, showing only marginal drop in erasure quality compared to CASteer applied on SD-1.4. With vectors computed on all denoising steps. CLIP score and FID metrics show that the general quality of generated images also remains high.

---

> ### Author Response · Authors · 2025-11-21
> **Continuing the rebuttal**
>
> **5. How does the method work if there is a mismatch between the number of steps used in steering vector extraction and during inference?**
>
> Note that for SDXL and SANA, we used steering vectors with only CA vector per CA layer computed using distilled versions of these models (SDXL-Turbo and SANA-Sprint, respectively). We  then applied them to generate images with full models with 20 or 30 denoising steps, using the same steering vectors for each CA layer across all diffusion steps. This does not depend on the numer of denoising steps used. For SD-1.4, as we have mentioned above, it is also possible to use one steering vector per CA layer across all denoising steps.
>
> **6. Concept erasure based on guidance or steering is susceptible to adversarial attacks via concept addition or subtraction. Is this robust to adversarial attacks based on concept arithmetic [1] ?**
>
> While an interesting question, **it lies outside the scope of our work**. None of the baseline methods we compare against evaluate robustness to adversarial concept arithmetic. Moreover, the referenced paper provides neither code nor a functioning project link, **making it infeasible for us to assess CASteer under these attacks**.
>
> **7. Results are only reported for few concepts (nudity, violence, snoopy, mickey). Missing erasure results on art styles (e.g., van gogh) or more abstract concepts (e.g., summer, mosaic style etc).**
>
> We thank the reviewer for this suggestion. We have added results on erasing art styles and abstract styles to both the main paper and the appendix. For art-style erasure, following SAFREE and ESD, we evaluate on two benchmarks: (i) five classical artists (Van Gogh, Picasso, Rembrandt, Warhol, Caravaggio) and (ii) five modern artists (McKernan, Kinkade, Edlin, Eng, Ajin: Demi-Human), with removal tasks focused on Van Gogh and McKernan. We report LPIPS scores and GPT-4o artist identification accuracy. Results **(Tab. 4 for SD-1.4; Sec. F–G for SDXL and SANA)** show that CASteer reliably removes target styles while preserving other styles.
>
> We also provide qualitative examples for Van Gogh, mosaic, and origami style erasure in **Fig. 6 (main paper) and Figs. 16, 17, 27, 28 (appendix)**.
>
> **8. Comparisons are only reported for SDv1.4 model. Missing comparisons to prior methods on more recent models such as SDXL, SANA or SD3.5**
>
> We are not aware of any prior work reporting metrics on SANA. For SDXL, we compare CASteer with the recently proposed SAFREE method in Sec. F.4 (appendix) across four nudity-erasure datasets: I2P, P4D, Ring-A-Bell, MMA-Diffusion, and UnlearnDiffAtk. CASteer achieves consistently superior performance on all benchmarks.
>
> **9. Paper can be better organized. Important ablations from supplementary need to be included in the main paper or the appendix instead of supplementary paper.**
>
> We thank the reviewer for raising this. We have moved the appendix from the supplementary to the appendix of the main paper. At the same time, we have added experiments in Style erasure (lines 465-475) in the main paper, in addition to a better justification of the choice of β and a new figure (Fig. 2).
>
> **10. How are the multiple prompts generated?**
>
> We describe how the prompts are generated for computing steering vectors across all tasks in **Sec. C** of the appendix.
>
> **11. Do the steering vectors generated for SDv1.4 work for SD v1.5 across different checkpoints of the same model?**
>
> Yes, steering vectors generated for SDv1.4 can be applied to SD v1.5. We provide results in this setting on all our main experiments (erasing nudity, harmful content, concrete concepts and styles) in Sec.M (SD-1.5 on Steering Vectors from SD-1.4) . Results show that CASteer applied to SD-1.5 with steering vectors computed on SD-1.4 results in substantial improvements similar to that of the SD-1.4 model across all tasks, while also preserving initial image quality.

---

### Official Review · Reviewer_ezmt · 2025-10-31

**Soundness:** 3
**Presentation:** 3
**Contribution:** 3
**Rating:** 6
**Confidence:** 3

**Summary:**

The paper proposed Cross-Attention Steering (CASteer), a training-free framework for concept erasure in diffusion models using steering vectors to influence hidden representations dynamically. Specifically, CASteer cleverly designs an algorithm for constructing steering vectors for new concepts and leverages these steering vectors to suppress unwanted image features without retraining. Extensive experiments demonstrate the effectiveness of the proposed method.

**Strengths:**

* The writing is fluent and logically coherent, exhibiting strong readability.
* The proposed method is highly efficient, requiring no training or fine-tuning while achieving excellent performance.
* The proposed method exhibits strong generalization and is applicable to various text-to-image models that incorporate cross-attention mechanisms.
* Comprehensive qualitative and quantitative experiments demonstrate the effectiveness of the proposed method.

**Weaknesses:**

* As shown in Figure 1, CASteer computes a steering vector for the output of every cross-attention layer at every timestep in the generation process and applies a correction. Could this be somewhat excessive? Would it be possible to experimentally analyze whether the number of corrected timesteps and CA layers can be reduced to improve efficiency?
* The reviewer examined the examples provided in the main paper as well as the prompt set in the appendix, and observed that the subjects used when constructing the prompts are relatively simple. This may lead to high variability in the generated outputs, which in turn could make the steering vector estimation unstable. For example, in Figure 1, when using “dog” as the subject, the model generates a brown dog as the positive sample and a white dog as the negative sample; this could cause the steering vector to incorrectly treat other attributes of the dog as part of the target concept. Would using more specific subjects yield a more stable and robust steering vector?
* It would be very helpful to provide some positive visual examples of erased abstract concepts, such as “Van Gogh style.”

**Questions:**

See 'Weaknesses'.

---

> ### Author Response · Authors · 2025-11-21
> **Rebuttal to Reviewer ezmt**
>
> We thank the reviewer for their positive review. We are delighted to see the reviewer calling our paper **exhibiting strong readability**, **highly efficient**, **achieving excellent performance**, **exhibits strong generalization**, and having **Comprehensive qualitative and quantitative experiments**. Below, we reply to the questions the reviewer raised.
>
> **1. CASteer computes a steering vector for the output of every cross-attention layer at every timestep in the generation process and applies a correction. Could this be somewhat excessive?**
>
> We study the effect of which CA layers to steer in **Sec. H.5 (appendix)**. We observe a trade-off: steering many layers can erase the target concept but also distorts global layout or identity (Fig. 30–31). Steering only the last few CA layers provides a better balance and effective concept removal with minimal disruption to other details (**Fig. 31**).
>
> Based on this, we hypothesize that texture-based concepts (e.g., styles) can be erased by steering only late CA layers, whereas concrete concepts (e.g., Snoopy) require deeper intervention. **We test this by steering only the last 36 CA layers of SDXL. Results (Tab. 26–27)** show that this setting successfully removes Van Gogh style while preserving non-target styles, but performs significantly worse for Snoopy erasure. This suggests that the optimal subset of CA layers is task-dependent.
>
> Finally, note that the clipping-free CASteer variant (“Injecting CASteer into model weights,” Sec. 3.3) adds no time or memory overhead regardless of how many layers are steered.
>
> **2. Would using more specific subjects yield a more stable and robust steering vector?**
>
> Individual prompt pairs can introduce unintended variations beyond the target concept (e.g., “a dog with Mickey” producing a white dog vs. “a dog” producing a brown dog). To reduce such pair-specific biases, we average CA-activation differences across many prompt pairs, which cancels out incidental variations and yields a steering vector that more cleanly captures the target concept.
>
> We assess sensitivity to prompt choice and prompt-set size in Sec. H.1 (appendix) using the Snoopy-erasure task with fixed β=2. Steering vectors are recomputed using different numbers of randomly sampled prompt pairs from Sec. C.1, and each is evaluated via the Snoopy-erasure experiment (Sec. 4.1). **We report the CLIP score for “Snoopy” and FID for non-target concepts. As shown in Fig. 29**, CASteer is robust to prompt selection, and performance stabilizes once ≥50 prompt pairs are used.
>
> **3. It would be very helpful to provide some positive visual examples of erased abstract concepts, such as “Van Gogh style.”**
>
> We thank the reviewer for this suggestion. We have added results on erasing art styles and abstract styles to both the main paper and the appendix. For art-style erasure, following SAFREE and ESD, we evaluate on two benchmarks: (i) five classical artists (Van Gogh, Picasso, Rembrandt, Warhol, Caravaggio) and (ii) five modern artists (McKernan, Kinkade, Edlin, Eng, Ajin: Demi-Human), with removal tasks focused on Van Gogh and McKernan. We report LPIPS scores and GPT-4o artist identification accuracy. Results **(Tab. 4 for SD-1.4; Sec. F–G for SDXL and SANA)** show that CASteer reliably removes target styles while preserving other styles.
>
> We also provide qualitative examples for Van Gogh, mosaic, and origami style erasure in **Fig. 6 (main paper) and Figs. 16, 17, 27, 28 (appendix)**.

---

> > ### Comment · Reviewer_ezmt · 2025-11-27
> >
> > Thank you. The authors’ response has addressed the concerns I raised, and I will maintain my positive score.

---

> > > ### Author Response · Authors · 2025-11-27
> > > **Thanks**
> > >
> > > We are happy to see that we solved the concerns raised by the reviewer, and that they continue supporting our paper being accepted.

---

### Official Review · Reviewer_vj4f · 2025-10-31

**Soundness:** 3
**Presentation:** 3
**Contribution:** 3
**Rating:** 6
**Confidence:** 4

**Summary:**

The paper introduces CASteer, a training-free method for concept erasure in text-to-image diffusion models. Given an input prompt and a concept prompt, CASteer guides image generation to follow the input prompt while avoiding visual content associated with the concept prompt. The method computes steering vectors from the difference between cross-attention outputs when prompting the model with and without the concept. These vectors are applied at inference to steer generation in all the cross attention layers of the diffusion transformers. Experiments show that CASteer significantly outperforms prior baselines and is able to remove both abstract (e.g. nudity, inappropriate content) and concrete concepts (e.g. Mickey, Spongebob) and to generalize across models such as Stable Diffusion v1.4, SDXL, and SANA.

**Strengths:**

1. The works presents a novel *training-free* approach to solve the problem of content erasure which improves over existing baselines both in term of performances and of practicality (no need to run an ad-hoc training).
2. I find the idea of using steering vectors to *remove* concepts rather than *adding* them very interesting.
3. The paper is well written and easy to follow.

**Weaknesses:**

I only have some minor concerns:

1. The *extension to multiple concepts* (L272–L276) is not experimentally validated. It would be useful to assess performance as the number of erased concepts increases and/or their individual steering vectors substantially differ from each other.
2. Equation 4 introduces *cosine-similarity weighting* between the text prompt and the erased concept; the paper does not analyze how crucial this weighting is in practice. An ablation could clarify whether simpler weighting performs comparably.
3. Some *figures* could be improved. Figures 2 and 4 could benefit from clearer labeling of rows/columns and spacing can be increased among the sub captions of Figure 3.

**Questions:**

1. How does CASteer perform when erasing multiple unrelated concepts or when steering vectors are different from each others?
2. How critical is cosine-similarity weighting? Could fixed or learned scalars achieve similar results?

---

> ### Author Response · Authors · 2025-11-21
> **Rebuttal to Reviewer vj4f**
>
> We thank the reviewer for their positive review. We are very pleased to see our paper being called **novel**, **interesting**, **well-written**, and **easy to follow**. Below, we reply to the **minor concerns** raised by the reviewer.
>
> **1) The extension to multiple concepts (L272–L276) is not experimentally validated.**
>
> This is an excellent question. We added a description of our multi-concept erasure procedure to the main paper (**Sec. 3.4**) and expanded quantitative results in the appendix (**Sec. M**).
>
> We propose two strategies:
>
> 1. **Averaging steering vectors.**
>    We create a single steering vector by averaging the individual vectors for each concept. This is the approach used for harmful-content erasure on the I2P dataset (Tab. 2), where we average vectors for hate, harassment, violence, self-harm, sexual, shocking, and illegal activity. This yields state-of-the-art performance on the T2I benchmark.
>
> 2. **Orthogonalized sequential erasure.**
>    For less related concepts (e.g., Snoopy and nudity), simple averaging can be suboptimal. Instead, we compute each concept’s steering vector, orthogonalize them via Gram–Schmidt, and then apply them sequentially using Eq. 4. Orthogonalization is done offline. Without clipping, each steering step has a matrix form (Eq. 5), so multiple-concept erasure can be implemented as one matrix multiplication, adding no inference-time overhead.
>
> **In Sec. M, we present results on erasing concept pairs (Snoopy + nudity, Snoopy + Mickey) using orthogonalized vectors**. CASteer successfully removes both concepts, achieving performance comparable to single-concept erasure.
>
> **2) Ablation on cosine similarity.**
>
> We thank the reviewer for the comment. Our use of dot-product weighting is motivated by the fact that the dot product between a CA output and a concept’s steering vector provides a good estimate of how strongly that concept is expressed in that token. We illustrate this in **Fig. 2**: for prompts containing Snoopy, the dot product is high precisely on the image tokens generating Snoopy; for prompts without Snoopy, dot-product values remain low across all tokens.
>
> We further validate this choice **in Sec. H.2 (appendix)**. We replace dot-product weighting with a constant multiplier across all tokens and layers and evaluate Snoopy erasure (Sec. 4) and nudity erasure (FID on COCO-30k prompts). With constant multipliers of 1 or 2, Snoopy is not erased; at multiplier 2, image quality degrades sharply, with FID increasing by 3.5× compared to CASteer.
> Finally, dot product is a natural measure of mutual information under the Linear Representation Hypothesis, and recent work shows that diffusion backbones contain intermediate subspaces with approximately linear semantic directions [A–D].
>
> [A] Kwon et al., Diffusion Models Already Have a Semantic Latent Space, ICLR 2023.
>
> [B] Park et al., Understanding the Latent Space of Diffusion Models Through the Lens of Riemannian Geometry, NeurIPS 2023.
>
> [C] Si et al., FreeU: Free Lunch in Diffusion U-Net, CVPR 2024.
>
> [D] Park et al., The Linear Representation Hypothesis and the Geometry of Large Language Models, ICML 2024
>
> **3) Some figures could be improved.**
>
> We have fixed the spacing for the figures.

---

> > ### Comment · Reviewer_vj4f · 2025-11-27
> >
> > Thank you for the additional experiments (especially the one on orthogonalization is very interesting!) and for your answers, they solved my minor concerns. I am happy to maintain my positive score and support acceptance.

---

> > > ### Author Response · Authors · 2025-11-27
> > > **Thanks**
> > >
> > > We are glad to solve the minor issues the reviewer raised, and happy to hear that they continue supporting our paper getting accepted.

---

### Official Review · Reviewer_ZNEe · 2025-11-01

**Soundness:** 3
**Presentation:** 2
**Contribution:** 2
**Rating:** 4
**Confidence:** 4

**Summary:**

This paper introduces CASteer, a training-free framework for erasing specific concepts from diffusion models. The core idea is to compute steering vectors that represent the direction of an unwanted concept in the model's cross-attention layers. During inference, these vectors are subtracted from the model's activations without requiring any model retraining. The authors did comprehensive empirical experiments to demonstrate the effectiveness of the proposed method across various concepts, prompts, and model architectures.

**Strengths:**

- a training-free framework using precomputed steering vectors to remove concepts from diffusion models.
- comprehensive empirical evaluations across various concepts and models.

**Weaknesses:**

- limited novelty regarding method, with strong assumptions on steering vectors’ linear compositionality.
- ad-hoc parameter selection - steering vector scaling hyperparameters are fixed empirically.
- no discussion of computational or memory trade-offs when constructing per-layer, per-step steering vectors.
- somewhat limited improvements, for example, CASteer with clipping only surpassing second-based model Receler by 1.42%, which is not significant.
- did not compare with unlearning baselines, such as [1,2].

[1] Wu, Yongliang, et al. "Unlearning concepts in diffusion model via concept domain correction and concept preserving gradient.

[2] Alberti, Silas, et al. "Data unlearning in diffusion models." arXiv preprint arXiv:2503.01034

**Questions:**

- the hyperparameter $β=2$ is used for all experiments. How does the method's performance and stability vary with this parameter, especially across different concepts or model architectures?
- the paper uses a large number of prompt pairs (50 for concrete, 196 for abstract concepts). How sensitive is the method to the quality and diversity of these prompt pairs? poorly chosen set of prompts could lead to an imprecise steering vector.
- for multi-concept erasure, how do you resolve conflicting steering directions? is orthogonality assumptions still valid?
- the results on using steering vectors for adding concepts (and style transfer) are rather limited, can you discuss why certain applications fails with the proposed method?

---

> ### Author Response · Authors · 2025-11-21
> **Rebuttal to Reviewer ZNEe**
>
> We thank the reviewer for the detailed review, and for **praising the comprehensiveness of our evaluation**. Below, we response to the reviewer's concerns, which included a significant number of new experiments.
>
> **1) On the lack of novelty of the method and the strong assumptions on steering vectors’ linear compositionality.**
>
> We are unaware of any prior work that uses non-trainable steering vectors directly to the cross-attention (CA) outputs of diffusion models for concept erasure. Existing approaches that exploit directional structure in diffusion activations (see “Utilizing directions in latent spaces” in Related Work) do not target CA activations. Moreover, many of these methods are architecture-dependent (e.g., requiring access to U-Net bottleneck layers) and training-intensive, limiting applicability to modern non-U-Net backbones such as SANA. Our experiments show that CASteer consistently outperforms these baselines (Tables 1, 2, 4; Fig. 4).
>
> Regarding the assumption of linear compositionality of steering vectors, recent work provides empirical support for this property: diffusion backbones exhibit intermediate subspaces with approximately linear semantic directions that modulate feature expressiveness [A–D].
>
> [A] Kwon et al., Diffusion Models Already Have a Semantic Latent Space, ICLR 2023.
> [B] Park et al., Understanding the Latent Space of Diffusion Models Through the Lens of Riemannian Geometry, NeurIPS 2023.
> [C] Si et al., FreeU: Free Lunch in Diffusion U-Net, CVPR 2024.
> [D] Park et al., The Linear Representation Hypothesis and the Geometry of Large Language Models, ICML 2024.
>
> **2) On the ad-hoc parameter selection.**
>
> We thank the reviewer for pointing out this point. We have added explanation of the choice of steering strength β to the sec.4 (Implementation details) of our paper, and also provided results on applying CASteer to the tasks of concrete concept erasure (Snoopy) and abstract concept erasure (nudity) with varying values of β in the appendix.
>
> The choice is motivated by the fact that with β = 2, the Eq. 5 (steering transform) becomes a Householder operator (reflection) of the Cross-Attention activation vector c across the hyperplane orthogonal to the steering vector s. This operation preserves L2-norm of the vector c, thus keeping relative and absolute values of all the information present in c that is orthogonal to s intact after transformation. We have added this discussion in lines 323-355. Furthermore, in appendix (see **Sec. H.3**), we also ablate the choice of β. While we reach the best results with β = 2, we show that β < 2 leads to lower level of concept suppression, still producing high quality images, enabling control on the level of concept erasure in practical applications. We also show that for SDXL and SANA models, values of β > 2 lead to stronger concept erasure, while also leaving general quality of images intact.
>
> **3) On discussing the computational and memory trade-offs.**
>
> Computing steering vectors for SDXL-Turbo/SANA-Sprint on 50 prompt pairs takes **1 minute on a single Nvidia V-100, and in the order of less than 10 seconds on an Nvidia-H100**. Note that this is much less than a typical optimization-based approach.
>
> In terms of memory, steering vectors computed by SDXL-Turbo for using in SDXL requires **an additional 6% of memory for SDXL, and an additional 8% of memory for SANA**. Finally, note that the CASteer version without clipping (see “Injecting CASteer into model weights” in **sec.3.3**) introduces no memory overhead during inference, as steering vectors do not need to be kept separately from model weights.
>
> **4) CASteer outperforms Receler by only 1.42%.**
>
> We strongly **disagree** with the reviewer that an 1.42% absolute improvement (**4.9% relative improvement**) over a very strong state-of-the-art baseline is not significant. Furthermore, we have also tested our approach across many tasks, such as concrete concepts erasure (fig.4), nudity erasure (tab.1), style erasure (tab.4). CASteer always shows consistent improvements across all these tasks, clearly outperforming all other methods in the tasks of  nudity and style erasure, and showing great performance in the task of concrete object erasure, while having a benefit of being able to erase implicitly defined concepts.

---

> ### Author Response · Authors · 2025-11-21
> **Continuing the Rebuttal**
>
> **5) Unlearning baselines.**
>
> We thank the reviewer for highlighting these works. We have incorporated Unlearning Concepts in Diffusion Models via Concept Domain Correction and Concept-Preserving Gradients into our comparisons (**Fig. 4** for concrete concept erasure, **Tab. 1** for nudity erasure), referring to it as **DoCo**. Using the authors’ publicly released checkpoint, we find that DoCo fails to erase nudity (**see Tab.1**). Additionally, we evaluate DoCo on implicit concept erasure and find that it often fails to do so. We have updated section related to erasing implicitly defined concepts in the main part of the paper (“CASteer is capable of erasing implicitly defined concepts” in **Sec. 4.1**) and in the appendix (**Sec. K**)
>
> Regarding Data Unlearning in Diffusion Models, this method targets memorization unlearning in Stable Diffusion, preventing reproduction of specific training images. It also requires a curated set of unlearning examples for each task, making it not directly applicable to general concept erasure.
>
> [1] Wu et al. "Unlearning concepts in diffusion model via concept domain correction and concept preserving gradient, AAAI 2025.
>
> [2] Alberti et al. "Data unlearning in diffusion models.", ICLR 2025
>
> **6) Hyperparameter β.**
>
> We thank the reviewer for pointing out this point. **We have added explanation of the choice of steering strength β to the sec.4 (Implementation details) of our paper**, and also provided results on applying CASteer to the tasks of concrete concept erasure (Snoopy) and abstract concept erasure (nudity) with varying values of β in the appendix (Sec. H.3).
>
> The choice is motivated by the fact that with β = 2, the Eq. 5 (steering transform) becomes a Householder operator (reflection) of the Cross-Attention activation vector c across the hyperplane orthogonal to the steering vector s. This operation preserves L2-norm of the vector c, thus keeping relative and absolute values of all the information present in c that is orthogonal to s intact after transformation. We have added this discussion in lines 323-355. Furthermore, in appendix (see Sec. H.3), we also ablate the choice of β. While we reach the best results with β = 2, we show that β < 2 leads to lower level of concept suppression, while still producing high quality images, enabling control on the level of concept erasure in practical applications. We also show that for SDXL and SANA models, values of β > 2 can lead to stronger concept erasure, while also leaving general quality of images intact.
>
> **7) How sensitive is the method to the quality and diversity of these prompt pairs?**
>
> For constructing steering vectors, we use prompt pairs differing only by the target concept, ensuring that differences in their CA outputs primarily encode that concept. Individual pairs may still introduce unrelated variation (e.g., “a dog with Mickey” yielding a white dog vs. “a dog” yielding a brown dog), so we average CA-activation differences across many prompt pairs. This suppresses pair-specific biases and isolates the target concept in the resulting steering vector.
>
> We evaluate sensitivity to prompt choice **in Sec. H.1** (appendix) using the Snoopy-erasure task with fixed β=2. Steering vectors are recomputed with varying numbers of randomly sampled prompt pairs from Sec. C.1. For each steering vector, we run the Snoopy-erasure experiment and record the CLIP score for “Snoopy” and the FID for non-target concepts. As shown **in Fig. 29**, CASteer is robust to prompt diversity, and performance stabilizes once ≥50 prompt pairs are used.
>
> **8) is orthogonality assumptions still valid?**
>
> This is an excellent question. We added a description of our multi-concept erasure procedure to the main paper (Sec. 3.4) and expanded quantitative results in the appendix (Sec. M).
>
> We propose two strategies:
>
> 1. **Averaging steering vectors.**
>    We create a single steering vector by averaging the individual vectors for each concept. This is the approach used for harmful-content erasure on the I2P dataset (**Tab. 2**), where we average vectors for hate, harassment, violence, self-harm, sexual, shocking, and illegal activity. This yields state-of-the-art performance on the T2I benchmark.
>
> 2. **Orthogonalized sequential erasure.**
>    For less related concepts (e.g., Snoopy and nudity), simple averaging can be suboptimal. Instead, we compute each concept’s steering vector, orthogonalize them via Gram–Schmidt, and then apply them sequentially using Eq. 4. Orthogonalization is done offline. Without clipping, each steering step has a matrix form (Eq. 5), so multiple-concept erasure can be implemented as one matrix multiplication, adding no inference-time overhead.
>
> In **Sec. M**, we present results on erasing concept pairs (Snoopy + nudity, Snoopy + Mickey) using orthogonalized vectors. CASteer successfully removes both concepts, achieving performance comparable to single-concept erasure.

---

> ### Author Response · Authors · 2025-11-21
> **Finalizing the rebuttal**
>
> **9) The results on using steering vectors for adding concepts (and style transfer) are rather limited, can you discuss why certain applications fails with the proposed method?**
>
> The primary focus of our paper is concept erasure. Results on concept addition and style transfer are included only as preliminary evidence that CASteer may generalize to other tasks, but a full exploration is left for future work.
>
> With regards to the possible limitations:
>
> **1. Concept addition.** There two primary challenges are: (i) Steering strength estimation: Unlike erasure, where strength can be inferred from the dot product between the CA activation and the steering vector, it is nontrivial to determine how much of a steering vector should be added to reliably introduce a concept. The required strength appears to vary across concepts and prompts. (ii) Spatial control: In many cases, a concept should appear only in a specific region rather than across the entire image. Our initial approach applies the steering vector to all image tokens, which nonetheless often yields localized insertions. We hypothesize that more precise targeting of specific spatial tokens could improve placement accuracy and reduce unwanted distortions.
>
> **2. Style transfer.** CASteer is designed to modify semantics during generation, whereas style transfer requires editing real images. To enable this, we use DDIM inversion [1] to map a real image to its latent noise, apply CASteer during the forward denoising process, and decode the result. This procedure inevitably leads to some loss of fine-grained details.
>
> [1] J. Song, C. Meng, S. Ermon. Denoising Diffusion Implicit Models. ICLR 2021.

---

### Author Response · Authors · 2025-11-21
**Common answer**

Dear Area Chair and Reviewers,

We genuinely think the reviewers for their work on helping us improve the paper. There were several nice suggestions and questions that made us dig deeper be it theoretically or empirically. While we apologize for the lengthy responses to Reviewer ZNEe and dKH9, we felt that it was needed to respond to each of their concerns, usually accompanied by one or more experiments, or a Theorem.

We made several additions and changes in the manuscript, all in blue text. In the individual rebuttals, we link to those changes and results. In particular, a non-comprehensive list of these changes is:

- Adding the appendix to the main paper, instead of in supplementary.

- Showing results for timing and memory performance.

- Doing all the asked ablations (e.g., different number of diffusion steps, evaluating the choice of β, sensitivity to prompt choice, cosine similarity, more visual results such as in Van Gogh style, steering only some layers, robustness to seed, mismatch on number of steps, steering multiple concepts, comparing with unlearning methods such as DoCo, etc). While we note that most of the reviewers already said that our method is **comprehensively evaluated**, we believe that the new set of experiments make the experimental setup even more stronger.

- We theoretically justified some of the choices be it in forms of Theorems  (see Appendix Sec. L for Theorem L.1 and Corollary L.1.1) or linking to the previous works (assumptions on steering vectors’ linear compositionality and orthogonality).

Considering the work done, we hope that the reviewers carefully revise the changes and the rebuttal. We are excited to further discuss with the reviewers in continuing to improve our paper.

---

### Meta-Review · Area_Chair_Vnz7 · 2026-01-12

**Summary:**

Reviews were initially mixed (6,6,4,4), and cited concerns about limited novelty, ad-hoc parameter selection, lack of discussion about method tradeoffs, limited improvement over baselines and a number of other things. The authors provided a comprehensive rebuttal that seems to have addressed all major points.

**Reviewer Concerns:**

Reviewers posed a number of concerns, but all seem to have been addressed in the rebuttal.

- Novelty
- Beta parameter selection
- Missing details about compute and memory
- Clarifying improvement over baselines
- Unlearning baseline comparisons
- Robustness to prompt variations
- Verify importance of cosine similarity
- Concerns about step mismatch

The only things that remain are confirmation from the reviewers that they are satisfied with the arguments about limited novelty and parameter selection.

**Reviewer Scores:**

Reviewer scores were mixed (6,6,4,4), but given the level of detail and the number of concerns addressed in the rebuttal, I would anticipate the scores of one or more marginally negative reviews to increase, leaving the submission's final rating at generally positive, e.g., (6,6,6,4) or (6,6,6,6).

---

### Decision · Program_Chairs · 2026-01-26

Accept (Poster)